# Medium-Level Architectures for Digital Twins: Bridging Conceptual Reference Architectures to Practical Implementation in Cloud, Edge and Cloud–Edge Deployments

Michel Fett [1,*], Marcel Kraft [1], Fabian Wilking [2], Stefan Goetz [2], Sandro Wartzack [2] and Eckhard Kirchner [1]

1 Institute for Product Development and Machine Elements, Technical University Darmstadt, Otto-Berndt-Straße 2, 64287 Darmstadt, Germany; marcel.kraft@stud.tu-darmstadt.de (M.K.); kirchner@pmd.tu-darmstadt.de (E.K.)

2 Engineering Design, Friedrich-Alexander-Universität Erlangen-Nürnberg, Martensstraße 9, 91058 Erlangen, Germany; wilking@mfk.fau.de (F.W.); goetz@mfk.fau.de (S.G.); wartzack@mfk.fau.de (S.W.)

* Correspondence: michel.fett@tu-darmstadt.de

**Abstract:** The integration of Digital Twins (DTs) is becoming increasingly important in various industries. This entails the need for a comprehensive and practical IT infrastructure framework. This paper presents a modifiable medium-level architecture that serves as a link between established reference architectures such as RAMI 4.0 and the pragmatic implementation of Digital Twins. The functionalities of an IT infrastructure are considered, and functional hardware and software components for fulfilling these are described. The proposed architecture is suitable for various deployment scenarios, including local, cloud and hybrid cloud–edge configurations. In order to improve the applicability of the medium-level architecture, a step-by-step procedure is also proposed, which helps to transfer the overarching requirements for a Digital Twin into a suitable IT infrastructure. Finally, the results are demonstrated by an exemplary application to a two-stage industrial gearbox.

**Keywords:** digital twin; cyber–physical system; RAMI 4.0; architecture; IT infrastructure

## 1. Introduction

Digital Twins (DTs) can be implemented in many different use cases and business models and offer great potential. Some of these potentials are preventive maintenance, supply chain management, resource planning and logistics [1]. Nevertheless, many companies, especially small- and medium-sized enterprises, do not utilise Digital Twins for their business model. One reason for this is that the development of Digital Twins requires consideration of different domains such as modelling, sensors and IT infrastructure, which makes the development a complex, interdisciplinary project. The IT infrastructure of a Digital Twin provides the computing power to simulate and calculate the models on which a Digital Twin is based. It also enables data exchange between the physical and digital space and between the individual models. The literature proposes examples and frameworks for IT infrastructures of Digital Twins. They may either concern an IT infrastructure on its own (e.g., [2]) or an IT infrastructure in cross-domain approaches (e.g., [3]). As part of a literature review, the articles on the creation of Digital Twins in general were analysed, and it was found that the IT infrastructure is only covered by about half as many articles as the models [4]. This may also be a reason why, according to an industry survey, 41% of participants desire more methodological support in the area of IT infrastructures [5]. In addition to the less frequent consideration, another reason for this may be that the IT infrastructure does not appear to be addressed in sufficient practical detail in the literature, which makes it difficult to apply. Reference architectures such as RAMI 4.0 [6] have a high degree of abstraction and are therefore challenging to apply directly. For this reason, this contribution bridges the gap between abstract reference architectures and individual

applications. The development of an IT infrastructure for Digital Twins is to be simplified in particular by creating a standardised understanding in the interdisciplinary development teams. Within the scope of this article, there is no ambition to provide further education for experienced software engineers, who undoubtedly play an important role. The following research questions will be addressed to reach this goal:

- RQ1: How can the gap between the abstract reference architectures and the highly individualised IT infrastructures of specific use cases be bridged?
- RQ2: Which steps must be taken to set up an IT infrastructure in practice?

In order to answer these research questions, this article first describes the foundations of Digital Twins (Section 2.1), reference architectures (Section 2.2), in particular RAMI 4.0 (Section 2.3), and IT infrastructures for Digital Twins (Section 2.4). Subsequently, a medium-level architecture is derived (Section 3) by the concretisation of RAMI 4.0 (Section 3.1) and abstraction through functional decomposition (Section 3.2) and then presented (Section 4). Furthermore, a process for applying the results is offered (Section 5), and the results are applied to an example (Section 6).

## 2. Fundamentals

In the following, the fundamentals necessary for this article are described. First, the basics of cyber–physical systems (CPSs) and the Digital Twin concept are presented. Furthermore, terms relating to IT infrastructure are clarified, and the RAMI 4.0 reference architecture is presented.

### 2.1. Fundamentals of Digital Twins

Mechanical systems, in which electrical or electronic components are integrated to extend the functionality, are known as mechatronic systems. These additional components can be sensors, actuators and embedded systems like microcontrollers. If one or multiple mechatronic systems are extended with components for communication, this is referred to as a cyber–physical system. This system is characterised by its connectivity with the Internet of Things and enables behaviour or properties to be adapted during operation [7].

On this basis, the concept of a digital counterpart to a physical system was first described in 2002 by Grieves in the context of product lifecycle management [8]. In 2010, this concept was called a Digital Twin in a NASA Roadmap [9]. A Digital Twin is a digital representation of a physical product. This product can be material or immaterial, but in the context of this article, the focus is limited to physical, technical products. These products are defined as physical twins and represent the physical counterpart of the Digital Twin. The digital representation is achieved through models which describe the behaviour of the physical twin. These models are fed with real-time data from the physical twin [10–12]. The concept of "real time" describes the relevance of the time at which a system creates an output [13]. This can be addressed by predetermined time intervals between the input and output. The length of these time intervals depends on the application. The data used for this can, for example, come from the operations, the control system or the environment of the physical twin. For this purpose, a suitable sensor system must be integrated. Subsequently, the data are used in models to perform calculations and simulations of the behaviour of the physical twin. If necessary, the data are stored and historical data are retrieved. The aim is to make predictions or conclusions about the behaviour of the physical twin using the Digital Twin [10,11].

Three key aspects are necessary to implement this Digital Twin concept: sensors, models and IT infrastructure. Sensors record the operating data of the physical twin; models describe the behaviour of the physical twin and can be used to calculate and simulate it. Furthermore, a suitable IT infrastructure enables bidirectional data exchange between the physical and digital world. This allows the calculation and simulation of the models. If the physical twin is already a CPS, the sensors, embedded systems and an initial communication interface already exist. To implement the Digital Twin concept, model-based data processing must be added, which requires a suitable IT infrastructure and

behaviour-describing models. If the physical twin is a mechatronic or solely a mechanical system, the communication interface or even the sensors and embedded system must be considered. This relationship between a mechanical, mechatronic and cyber–physical system and the Digital Twin concept can be seen in Figure 1.

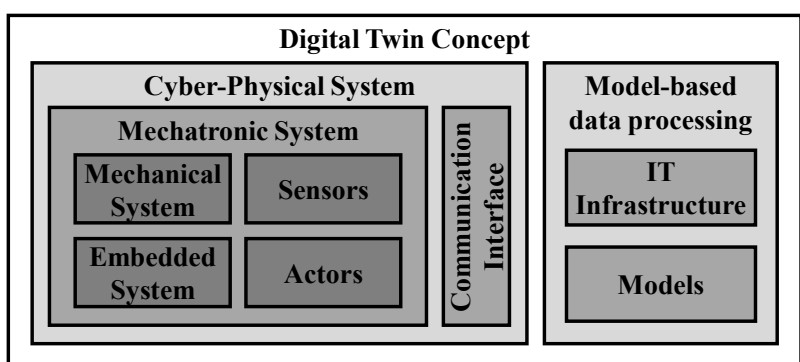

**Figure 1.** Relationship between the mechanical, mechatronic and cyber–physical system and the Digital Twin concept.

The results of the calculations or simulations by the models are fed back into the physical space, creating a bidirectional connection between the digital space and the physical space. There are different ways of utilising the results. With an Informational Digital Twin (IDT), the results are displayed to the user, for example, via a user interface on a screen. The Supporting Digital Twin (SDT), on the other hand, has an additional logic that enables the translation of the results into specific recommendations for action that it can offer the user. These recommendations for action are also made accessible via a user interface. A special case is the Autonomous Digital Twin (ADT). In this case, the results are used to directly adapt the physical twin's operation [14]. This may require interfaces to its control system or an integration of additional actuators. In this special case, a bidirectional connection is created between physical and digital space and additionally between the physical and Digital Twin. Figure 2 shows the DT concept with the bidirectional connection between physical and digital space/twin.

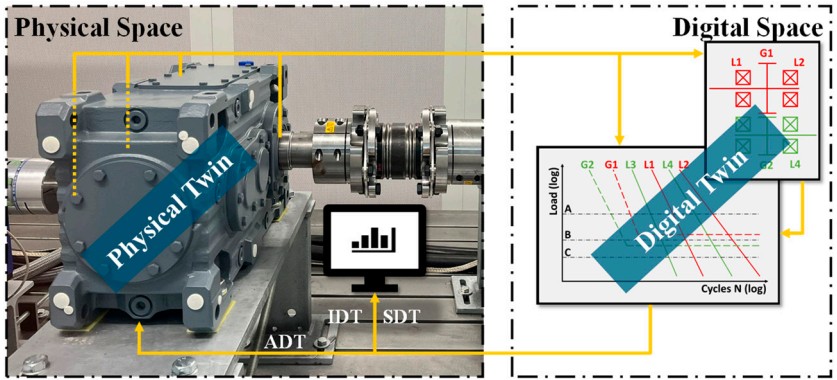

**Figure 2.** DT concept with the bidirectional connection between physical and data space/twin.

### 2.2. Fundamentals of Reference Architectures

The primary objective of this article is to support the creation of an IT infrastructure required for a Digital Twin. This enables data exchange and provides the necessary storage and computing capacity for the simulation or calculation of the models on which a Digital Twin is based. In order to create a Digital Twin and its IT infrastructure, reference architectures can be used. In the context of IT infrastructures and reference architectures, the terms IoT, architecture and framework are also frequently used. In order to create a uniform

understanding within the context of this article, these terms are defined and differentiated from one another in the following.

The Term IoT is an acronym for Internet of Things. Various authors have analysed the literature on the IoT and translated it into definitions [15–17]. According to Gubbi et al., the definition of the IoT for smart environments is the "Interconnection of sensing and actuating devices providing the ability to share information across platforms through a unified framework, developing a common operating picture for enabling innovative applications. This is achieved by seamless ubiquitous sensing, data analytics and information representation with Cloud computing as the unifying framework" [15]. Dorsemaine et al. describe the IoT as a "Group of infrastructures interconnecting connected objects and allowing their management, data mining and the access to the data they generate". They describe connected objects as "Sensor(s) and/or actuator(s) carrying out a specific function and that are able to communicate with other equipment. It is part of an infrastructure allowing the transport, storage, processing and access to the generated data by users or other systems" [16]. Atzori et al. understand the IoT as "a conceptual framework that leverages on the availability of heterogeneous devices and interconnection solutions, as well as augmented physical objects providing a shared information base on global scale, to support the design of applications involving at the same virtual level both people and representations of objects" [17]. In line with this, but simplified, the Gartner glossary describes the IoT as "the network of physical objects that contain embedded technology to communicate and sense or interact with their internal states or the external environment" [18]. In the context of this article, the term Internet of Things is understood to mean the interconnection of physical objects that are equipped with sensors, actuators and communication interfaces. The aim is to create a holistic picture of all connected systems with the possibility of joint interaction, management and data acquisition.

The term "infrastructure" is used in many different domains, so it is challenging to give a general scientific definition. However, one definition of non-technical dictionaries describes an infrastructure as "the basic structure of an organization or system which is necessary for its operation [. . .]" [19]. In the same dictionary, infrastructure in the context of IT is described as "the equipment, software, etc. that a computer system needs in order to operate and communicate with other computers" [19]. In line with this, the definition in the Gartner glossary is that an „IT infrastructure is the system of hardware, software, facilities and service components that support the delivery of business systems and IT-enabled processes" [20]. In the scope of this contribution, these definitions are summarised to that an IT infrastructure describes the components as well as the structure of hardware, software, facilities and service components of a system, which are necessary to fulfil its intended functionality.

The term "architecture" is described by Wang et al. as "[. . .] a unified structure for the purpose of implementing a technology. It can be used to decompose technology into key elements and help to integrate them into existing or new ecosystem with minimal efforts" [21]. According to Alan Clements, in the context of computer hardware, the "architecture describes the internal organization of a computer in an abstract way; that is, it defines the capabilities of the computer and its programming model" [22]. He also emphasises that "you can have two computers that have been constructed in different ways with different technologies but with the same architecture" [22]. Hennessy and Patterson point out that in the process the "instruction set design, functional organization, logic design, and implementation" [23] must be considered. In the scope of this contribution, an architecture describes an abstract, unified, internal structure of a (computer) system respective to its IT infrastructure, which is decomposed into key elements. The use of this architecture to implement the system is optional and represents the transition to the reference architecture. According to the IoT Consortium, a reference architecture is "[. . .] a type of architecture description that provides a set of constraints and guidance based on a set of related systems. A reference architecture contains information identifying the fundamental architecture constructs and specifies concerns, stakeholders, viewpoints,

model kinds, correspondence rules and conditions of applicability" [24]. Together with the considerations on the usability of an architecture provided above, the understanding in the context of this article is the following: a reference architecture is an architecture that is intended to assist in the implementation of a system with reduced effort. In addition to the description of the architecture itself, the reference architecture contains further information and guidance to enable the architecture to be used for the specific implementation. The result is not predefined; under the same conditions, different systems can be achieved.

The term "framework" is also used in the literature. Johnson and Foote understand this to be a "[...] a semi-complete application. A framework provides a reusable, common structure to share among applications. Developers incorporate the framework into their own application and extend it to meet their specific needs. Frameworks differ from toolkits by providing a coherent structure, rather than a simple set of utility classes" [25]. Similarly, Wolfgang Pree describes a framework as "a reusable semi-finished architecture for various application domains. [...] not only single building blocks but whole software (sub-)systems including their design can be reused" [26]. According to Dirk Riehle, a framework is a reusable design implementation approach that contains interrelated classes and rules [27]. In the context of this article, a framework is understood to be a semi-finished solution that consists of a reusable structure with reusable components and (sub-)systems. It can either be used alone or in combination with existing systems or other frameworks and can be modified or supplemented for the specific task. Compared to the (reference) architecture, the framework has a lower degree of abstraction and can be reused in a more direct way.

This article focuses on the reference architectures. The literature contains a number of reference architectures that can be used in the context of Digital Twins. Some examples of this are as follows:

- RAMI 4.0—Reference Architectural Model Industry 4.0 (DIN SPEC 91345) [6];
- IMSA—Intelligent Manufacturing System Architecture [28,29];
- IIRA—Industrial Internet Reference Architecture [24];
- Internet of Things (IoT)—Reference Architecture (ISO/IEC 30141) [30];
- Automation Systems and Integration—Digital Twin Framework for Manufacturing (ISO 23247) [31];
- 5C Model for Cyber–Physical Systems [32];
- Information technology—Open System Interconnection (ISO/IEC 7498) [33];
- Digital Twin Reference Architecture [34];
- Edge Computing Reference Architecture [35].

Furthermore, approaches like SAMM (previously known as BAMM) in the context of the open manufacturing platform [36] or Eclipse Ditto [37] could be utilisable.

All these concepts have many similarities and overlaps and present a template for an architecture, framework or IT infrastructure in some form. At the same time, all approaches aim to achieve a certain degree of universality in their respective areas of application. In order to achieve this degree of universality, all approaches have a very high level of abstraction, which in turn hinders their direct applicability. In order to apply these abstracts approaches, expert knowledge in the field of IT infrastructures is required on the one hand but also an in-depth understanding of the individual system on the other. Interdisciplinary development teams, especially in small companies with less experience, are therefore offered little help in creating an IT infrastructure for Digital Twins.

This article will focus primarily on RAMI 4.0. For this reason, it is described in more detail in the next section.

### 2.3. RAMI 4.0 (Reference Architectural Model Industry 4.0)

RAMI 4.0 is an acronym for "Reference Architectural Model Industry 4.0" (German: "Referenzarchitekturmodell Industrie 4.0") and is part of the German government's project to digitalise industrial production [6]. Concepts can be placed and described in the framework using three dimensions, which are represented in three axes: the architecture axis, the

life-cycle and value stream axis and the hierarchy axis, cf. Figure 3. This makes it possible to divide concepts into smaller, manageable sections and identify the need for development.

The architecture axis consists of six layers, which are described below. The business layer describes the business view, such as the mapping of business models and processes, as well as organisational and financial boundary conditions. The underlying functional layer deals with the function of a concept in its role in the overall system with the aim of being able to offer the business case. The information layer describes the data that the concept requires in order to fulfil its functionality. This includes, for example, the formal description of the models and consistent integration of data. The communication layer contains the data exchange of the connected components in the concept. Requirements such as I4.0 (Industry 4.0) conformity of data access must be considered. The underlying integration layer represents the transition from the physical to the digital world ("information world"). The infrastructure required to realise the function of the concept (resources such as hardware, software and documents) is also described here as well as interfaces such as the human–machine interface (HMI), sensors and actuators. The lowest layer is the asset layer and represents reality. This refers in particular to the totality of the real existing assets, people and interactions with the environment. The interface to the integration layer above is also arranged in this asset layer.

The second axis represents the life cycle and value stream. In DIN SPEC 91345, which introduces RAMI 4.0, IEC 62890 [38] is used for this purpose, which divides this dimension into development and maintenance/usage of the type, followed by production and maintenance/usage of the instance.

The last axis is the hierarchy axis, which describes the scope of the concept being addressed. This is based on the reference architecture for a factory in accordance with DIN EN 62264-1 [39] and DIN EN 61512-1 [40] and supplements these. The categories in this dimension are product, field device, control device, station, work centres, enterprise and connected world.

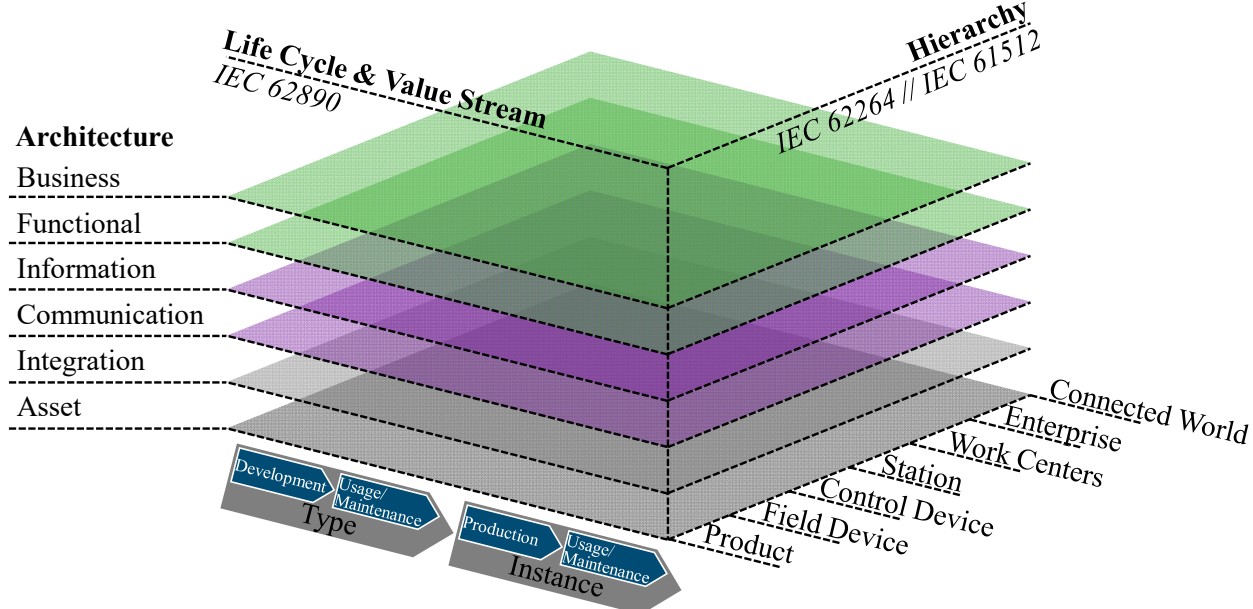

**Figure 3.** Reference Architectural Model Industry 4.0 (RAMI 4.0) [6].

On the architecture axis, the foundation is the asset that serves as the starting point for all other layers. An asset might represent different entities, such as a production system, an individual machine or an assembly within a machine. It must be uniquely identifiable, e.g., by a serial number or similar. In order to convert the asset into an I4.0 component, which represents a clearly identifiable participant in an I4.0 system, the asset is combined with a corresponding asset administration shell (AAS). The AAS is the virtual and active

representation of the asset in an I4.0 system. It contains features and functions of the asset from various domains in delimited sub-models but can also continuously collect and store information about the asset [6].

The AAS and the Digital Twin concept are similar. Both concepts describe a virtual representation of a physical counterpart. Abdel-Aty et al. [41] have analysed a number of studies on this topic and found that about half of the authors use the terms AAS and DT synonymously. The other half use the AAS as an implementation of a DT or as an information model for it. They consider the synonymous use of the two terms to be not appropriate, as the functional scope of Digital Twins and the associated requirements and criteria are not completely covered by the AAS [41]. This concerns, for example, the bidirectional connection between the Digital Twin and the physical twin and the real-time calculation or simulation of behaviour for the purpose of making predictions or drawing conclusions. This delimitation of the two concepts is agreed upon in the context of this contribution and will be adopted.

### *2.4. IT Infrastructures for Digital Twins*

The IT infrastructure of a Digital Twin is highly dependent on the use case. Depending on factors such as the type of physical twin, the scope and level of detail of the represented aspects, the desired range of functions, speed, data security and acceptable costs, considerable differences can be observed. These differences can be, for example, different versions of individual aspects but can also go so far that the resulting IT infrastructures are so dissimilar that they can hardly be compared. Due to this broad spectrum of possible IT infrastructures, they will not be discussed in detail in this article. Instead, reference is made to the existing literature reviews, which were also used to draw up some of the findings of this article.

Newrzella et al. [34] compare various Digital Twin architectures in terms of their functionality. They derive the functional elements of the physical entity, integration element, data management and information element, modelling and simulation element and the decision and user interfacing element. Fett et al. [4] have analysed the literature on the development of Digital Twins. In the domains of IT infrastructure, they identified the distinction between a physical space, a data space and a communication interface in between. Further elements of the IT infrastructure can be found in these supergroups. In the same publication, Fett et al. [4] consider step-by-step procedures for the development of Digital Twins, cyber–physical systems and product–service systems. The IT infrastructure is considered in individual steps of holistic approaches. One example of this is the procedure according to Nogueira de Andrade et al. [42] in which, among other things, the communication between models is established. Psarommatis and May [3] suggest selecting a suitable technology after determining the requirements and then carrying out performance testing. Jensen et al. [43] recommend addressing the computation problem, synthesising software code and finally verifying, validating and testing the entire system.

### 3. Derivation of Medium-Level Architectures through Externalisation of Knowledge

The reference architectures, of which RAMI 4.0 is a representative, have a high degree of universality. In order to achieve this, however, a high degree of abstraction is also necessary. Direct applicability and transfer into practical use is possible but requires a high degree of system understanding and experience. The creation of generally valid templates for specific and directly applicable low-level architectures is difficult due to their strong dependence on the specific use case. One approach to reducing this conflict is to introduce medium-level architectures. These can serve as a funnel from the abstract and universal high-level architectures to the use-case-specific, concrete and practical architectures.

For this purpose, the theoretical and abstract model of RAMI 4.0 is merged with the Digital Twin concept and thus concretised for this application. Furthermore, the empirical approaches described in Section 2.4 are abstracted. These are broken down functionally, and use-case-specific requirements are formulated. The results of the concretisation and

abstraction are merged into a medium-level architecture for Digital Twins. This procedure is shown schematically in Figure 4 and is described in more detail in the following sections. The concretisation and abstraction correspond to deduction and induction for the externalisation of knowledge [44,45]. The resulting medium-level architecture shown centrally in Figure 4 is described later in Section 4.

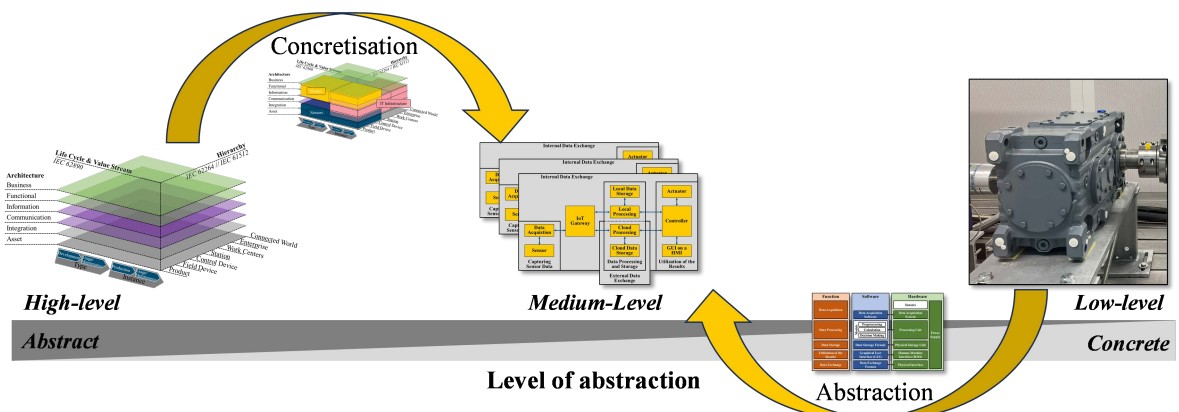

**Figure 4.** Derivation of a medium-level architecture through concretisation and abstraction.

### 3.1. Merging RAMI 4.0 and the Digital Twin Concept (Concretisation)

During concretisation, the concept of RAMI 4.0 is merged with the Digital Twin concept and thus brought closer to an application. This utilises the idea that RAMI 4.0 is intended to classify concepts according to three dimensions (architecture, life cycle and hierarchy) and divide them into smaller, manageable sections and identify the need for research. To this end, the Digital Twin concept is divided into the three domains of sensors, models and IT infrastructure, and these are then categorised in the RAMI 4.0 in the following, cf. Figure 5.

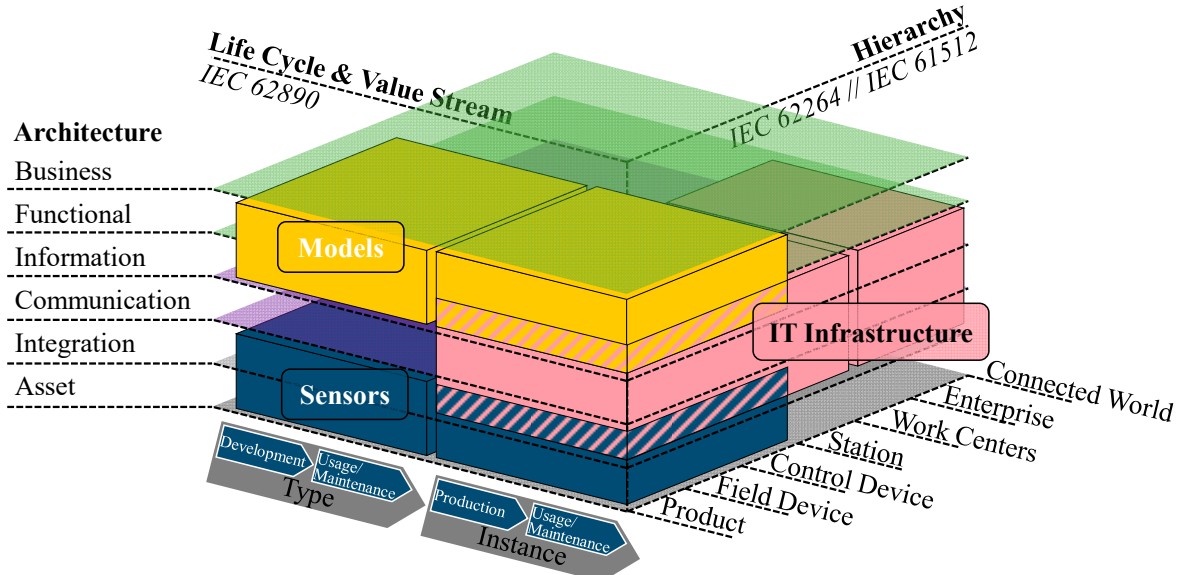

**Figure 5.** RAMI 4.0 with the domains (models, IT infrastructure, sensors) of a Digital Twin.

### 3.1.1. Models

The behaviour of the physical twin (asset) is depicted in the models, which is then calculated or simulated by a Digital Twin. In addition to calculation or simulation, models designed for this purpose can also be used to enable the preprocessing of the data or independent decision making by a Digital Twin. Although the models often represent

the behaviour of the asset, they are not assigned to the asset layer of the architecture axis. This is because the models contain the core functionality of Digital Twins and are therefore located in the functional and information layer with regard to the architecture axis of the RAMI 4.0. In special cases, the business layers may also overlap.

The consideration of Digital Twins in the scope of this article is limited to technical systems. These can be individual components, assemblies or entire systems. In the RAMI 4.0 hierarchy axis, this refers to the consideration of products, field devices and control devices. Higher-order systems such as entire stations and work centres are no longer considered. However, it should be noted that descriptions of shop-floor Digital Twins or even city Digital Twins can be found in the literature [1,46,47]. The insights gained there may also be transferable to higher hierarchy levels such as stations or work centres.

With regard to the life cycle, models and their development can be divided into two categories. The first category includes the creation of the models during development and the maintenance/usage of the superordinate type. Here, digital prototypes are assembled to form a digital master, which describes the behaviour of the type. During the instantiation/production of the respective instance, this digital master is linked with the digital shadow, which contains the specific information of the type, and forms a Digital Twin [10]. The second category describes the creation of the models of a Digital Twin at a point in time when the physical twin already exists. This corresponds to the production and maintenance/usage of the instance. This procedure is equivalent to retrofitting. Appropriate models are created for the intended use case. These can be created from scratch or be derived from models that were used in advance during product development.

In Figure 5, the domain of the models is represented with yellow cubes.

### 3.1.2. IT Infrastructure

An IT infrastructure contains hardware and software necessary to enable the exchange of data in accordance with the requirements placed on a Digital Twin. It enables the models to be calculated or simulated and the data to be stored and retrieved in a database. The description of the data and their exchange, as well as components such as hardware and software, places the IT infrastructure in the information layer, communication layer and integration layer with regard to the architecture axis of the RAMI 4.0. This requires overlaps with the models in the information layer and overlaps with the sensors in the integration layer. These overlaps represent interfaces to these domains.

With regard to the hierarchy axis in the RAMI 4.0, an IT infrastructure can be used to establish a connection within the product up to a connected world depending on the use case. This spectrum can be divided into two categories. Products, field devices, control devices and stations are typically connected via a single local network. Everything beyond this, such as work centres, enterprises or the connected world, requires a transition from the local network to at least one external network.

With regard to the life cycle axis of RAMI 4.0, consideration of the IT infrastructure during the development and maintenance/usage of the superordinate type is of limited value, as there is no direct influence on the physical product. The IT infrastructure only needs to be taken into account with the production and maintenance/usage of the respective instance.

The domain of the IT infrastructure is represented in Figure 5 with red cubes. The interactions between the IT infrastructure and the models and sensors are highlighted by red–yellow and red–blue hatching, respectively.

### 3.1.3. Sensors and Other Modifications of the Physical Twin

Sensors in and around the physical twin are required to feed the models within a Digital Twin with real operating data. These can be conventional sensors but also sensing machine elements or soft sensors [11,48]. Control data or higher-level enterprise manufacturing systems can also be used. The sensors represent a hardware interface to the physical twin. Other representatives of this interface are actuators, which are not used to detect the operation but actively manipulate it. Both sensors and actuators usually come

with a controller as an adapter. In addition, an HMI may be necessary as an interface to the user. The sensors, actuators, controller, HMI and associated components are located in the architecture axis of the RAMI 4.0 on the integration layer. Generally, the integration of these components requires modifications to the physical product, which is located on the asset layer together with the user and the environment. The sensors and other modifications to the physical system must therefore be considered equally for the integration and asset layers. In some reference architectures (e.g., [32,34]), initial preprocessing of the data is also carried out in the same layer as the sensor system. In the context of this article, this is explicitly not conducted; all data processing in the various stages is carried out in the models in the functional layer in order to have everything in one place.

Analogous to the models, the sensors in the context of this article are integrated primarily into products, field devices and control devices regarding the hierarchy axis of RAMI 4.0. Beyond this, systems for area monitoring such as cameras, GPS or similar are necessary. Initial approaches to this can also be found in the literature [1,47], but this will not be the subject of this article.

The consideration of the sensors and other modifications of the physical twin with regard to the life cycle axis is also analogous to the models and can be divided into two categories. The first category covers the selection and integration of sensors and other modifications during development and maintenance/usage of the superordinate type. This involves intervening in the product development process as early as possible in order to avoid undesirable interactions with the overall system and instead utilise synergy effects. The second category describes the integration of sensors and other modifications for an existing product after retrofitting [49].

In Figure 5, the domain of the sensors is represented with blue cubes.

### 3.2. Abstracting of Empirical IT Infrastructures (Abstraction)

The applications described in Section 2.4 are abstracted and broken down in terms of their functionalities. Through this increase in the level of abstraction, the universal applicability is increased as well. It is also possible to derive overarching use case requirements from this state.

#### 3.2.1. Functional Decomposition of the IT Infrastructures

An IT infrastructure must fulfil certain functions. In order to reduce the abstraction level of the reference architecture to a less abstract level and thus improve usability, it is decomposed functionally. The individual functions are then assigned to software and hardware components that are necessary to fulfil the functions. On the hardware side, the power supply must be taken into account for each component. However, this is not mentioned each time in the following descriptions.

Figure 6 shows the functional decomposition of the architecture. The functions are shown in orange, and the respective software and hardware required for them are shown in blue and green. The sensors and models are also considered in this overview, but the focus here is on the interfaces between them and an IT infrastructure. The models and sensors play such a central role that they need to be considered separately in more detail.

Data acquisition: The operating data of the physical twin are recorded by sensors. The sensors and the selection and integration of these represent a separate domain in the development of a Digital Twin, which must be considered in parallel with the IT infrastructure. In order to maintain the focus of this contribution, it is therefore referred to preliminary work in this domain for further information [50]. As the interactions between the two domains cannot be ignored, the sensors are nevertheless listed here as part of the hardware in the physical space. The sensors are physically connected to a data acquisition system. This system is used to supply the sensors with an input voltage and to measure and digitise the output voltage. For this purpose, the data acquisition system is operated with system-specific software. One example is the HBM QuantumX universal measuring amplifier in combination with the in-house software Catman. Suitable combinations of

hardware and software may not be commercially available for all tasks. Depending on the application, it may be necessary to use/modify/create own hardware using custom-made circuit boards or microcontrollers such as a Raspberry Pi or an Arduino. This must then be equipped with suitable software, which must be written, for example, in Python, C, C++, C# or modifications of these.

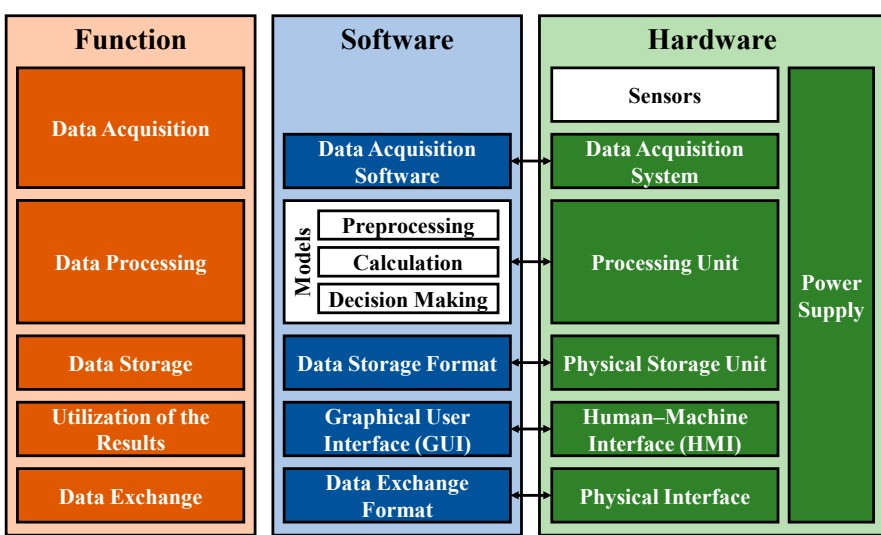

**Figure 6.** Overview of the functions and the associated software and hardware components of the IT infrastructure.

Data processing: The core of a Digital Twin consists of calculation models and/or computational models that represent the behaviour of the physical twin. Similar to the sensors, the models represent one of the key domains for the development of Digital Twins, which must be considered in parallel with the IT infrastructure. Reference is therefore also made here to further literature [4]. For the IT infrastructure, it is important to know that the models fulfil various tasks, such as preprocessing, calculation or decision making. Depending on the requirements, they can vary in complexity and computational intensity. The degree of computational intensity has an impact on the hardware-side processing unit and how powerful it needs to be. Examples of processing units are microcontrollers such as a Raspberry Pi or Arduino or even commercially available laptops or tower PCs. Local servers can also be used for more computationally intensive tasks. These examples are typically used locally at the location of operation. However, the hardware does not necessarily have to be located on site. With cloud computing, the computing hardware is located at centralised computing centres operated by cloud providers such as AWS or Google Cloud. The computing work is outsourced to these providers, which requires suitable interfaces.

Data storage: Depending on the use case, different data have to be stored. These can be, for example, the raw data of the data acquisition or intermediate or final results of the various calculations. There may also be use cases that do not require any data storage at all. For data storage, the data must be converted into a suitable data storage format by the software. Depending on the requirements, a balance must be found between memory requirements, read and write speed and accessibility. Examples of common data storage formats are SQL, noSQL [51–53] or CSV. There are various hardware solutions for data storage, which are based on different physical effects. These can be hard disk drives (HDDs), solid-state drives (SSDs) or flash drives, such as those often found in USB sticks. CDs or DVDs can also have advantages. As with data processing, data storage can also be centralised on servers of cloud providers.

Utilisation of the results: To utilise the results, they can either be presented to a user or used to adapt the physical twin or its operation [14]. The presentation to a user takes place on a physical human–machine interface (HMI), usually through the use of displays. These

can be standard commercial monitors as well as modern virtual reality (VR) or augmented reality (AR) lenses. The use of wearables such as a smart watch is also possible. On the software side, a graphical user interface (GUI) is used on these displays. The results of the calculation or simulation can simply be displayed as a string or visualised in the form of plots or graphs. Depending on the application, the use of three-dimensional digital models is also possible. Other, non-visual, interfaces such as acoustic or haptic interfaces are not widely used in the context of Digital Twins but can also have advantages under certain circumstances.

Alternatively, the results can also be used to adapt the physical twin or its operation [14]. This requires controllers that convert the data into control commands and actuators that manipulate the physical twin. For this purpose, either the actuators can be used, which are already necessarily integrated for the operation of the physical twin, or control actuators specially integrated for this purpose.

Data exchange: There is permanent data exchange between all the software and hardware components described. The interface can be different for each combination of components. A distinction can be made between an internal data exchange between the local components and an external exchange, which takes into account the communication of external networks such as cloud connections. For data exchange between the software components, communication protocols such as OPC UA or MQTT are widely used [54]. Between the physical components, physical interfaces are used. If several components are already part of a superordinate system, such as a laptop, the internal physical connection is realised via circuit boards or permanently soldered cables. In more modular systems, USB, SATA, PCI, LAN or other BUS standards are used. Wireless connections such as Wi-Fi, 5G or Bluetooth can also be applied. Depending on the application and architecture type, security concepts in the area of hardware and software must be taken into account for the data exchange.

A particular role in data exchange is played by the IoT communication. An IoT communication protocol (OPC UA, MQTT...) is used for this purpose, which ensures communication between the embedded systems (data acquisition device and control unit) and the software application (e.g., IoT platform) on which the models are located. If the embedded system and the software application are not compatible for a common IoT communication protocol, an additional IoT gateway must be implemented on the software side. The IoT gateway translates between different protocols [55].

### 3.2.2. Requirements for an IT Infrastructure of Digital Twins

Depending on the use case, stakeholders may have a variety of requirements with different specifications for an IT infrastructure. Due to the use case specificity, it is not possible to present these in their entirety here. Instead, the focus lays on the key requirements, which can be used to make an initial specification of the necessary IT infrastructure.

An IT infrastructure is used, among other things, to enable the calculations and simulations of the models on which a Digital Twin is based. This requires it to provide the necessary computing power. Depending on the application, the required computing power can be very volatile and at times exhibit strong performance peaks [56]. If a Digital Twin or its models are to be expandable, this must also be taken into account in terms of the scalability of the computing power.

Real-time capability is an essential aspect of Digital Twins. The term "real-time capability" essentially only describes the fact that data are collected and calculations are performed at regular intervals. The required time intervals between these calculations must be taken into account as part of the requirements for an IT infrastructure.

The data volumes to be processed also result in requirements for an IT infrastructure. On the one hand, this concerns the bandwidth for transferring large volumes of data in a short period of time and, on the other, the storage capacity for data volumes over a long period of time.

Another key issue is data security and data sovereignty. The loss or theft of data may be avoided. The corruption of data or the unauthorised input of incorrect data from outside also represent security risks [57]. Conscious access control must be taken into account. One possible approach for this could be the encryption of transferred data and algorithms [57–59].

There may also be requirements regarding the availability of a Digital Twin or the functionality provided by it. In particular, the consequences of long-term but also short-term unavailability must be taken into account, which may be security-relevant. This also includes the dependence on the service of the Internet provider.

For some use cases, it is relevant to be able to use a Digital Twin regardless of location. This must be ensured through appropriate accessibility [56].

In addition to these technical aspects, economic factors and, in particular, the costs must be taken into account. These include investment costs for hardware and software on the one hand but also ongoing costs for subscriptions and services such as cloud services. Personnel costs for service technicians or IT experts must also be taken into account. These can be company employees or external service providers or consultants.

Energy consumption is another cost factor. In addition, the efficient use of energy has increasingly come into focus, especially in recent years, due to the sustainable and environment-friendly usage of systems.

## 4. The Medium-Level Architectures

The concretisation of abstract theories and abstraction of specific applications make it possible to derive an adaptable medium-level architecture, which has a decent level of applicability due to its moderate degree of abstraction. The requirements and functionalities can vary depending on the application; therefore, the medium-level architecture needs to be flexible. In the following, three medium-level architectures, which are based vaguely on the work of Redelinghuys et al. [60,61], are presented. These are to be understood as a template for sorting the large number of possible solutions. In practice, modifications and mixed forms are possible.

The first elements are used for the data acquisition and are identical for all three architectures and describe the CPS or the interface to it. The mechatronic system and therefore also the sensors and actuators are not part of the IT infrastructure. Nevertheless, they are briefly discussed here, as the interface to them represents a key part of the IT infrastructure. The first element contains the sensors, which are used to collect operating data from the physical twin, and is therefore the direct interface to the physical twin. The second element contains the embedded system (consisting of the data acquisition device and the control unit) to which the sensors are connected. The communication capability is established through a communication interface. This could be an integrated network interface or an extension of the embedded system by means of a network card [62].

### 4.1. Local Architecture

The basis for the local architecture is local computing. This describes the decentralised collection, processing and storage of data in the local network close to the source where the data are produced [63]. For large or complex systems and the resulting high demands on performance, it is advisable to separate the components of the local architecture (local processing and local data storage) and adapt them separately to the requirements. Small or less complex systems with lower performance requirements are a special case. In this case, the components are not separated but combined and implemented on a single end device. This can take the form of a laptop or tower PC, for example.

Following on from the first elements of the architecture, which form the CPS or the interface to it, the adjacent element contains the communication protocol which ensures communication between the embedded systems and the element for local data processing. This element is located on a more powerful computing unit in the same local network and enables simulations and calculations by providing simulation software. The database is

located on a local server. This is used to store data and take historical data into account for diagnostic purposes [52]. The last element is the human–machine interface. The results of the data processing are sent to a client application, which can be used by the user to interact with the DT. Figure 7 shows a schematic representation of the local architecture.

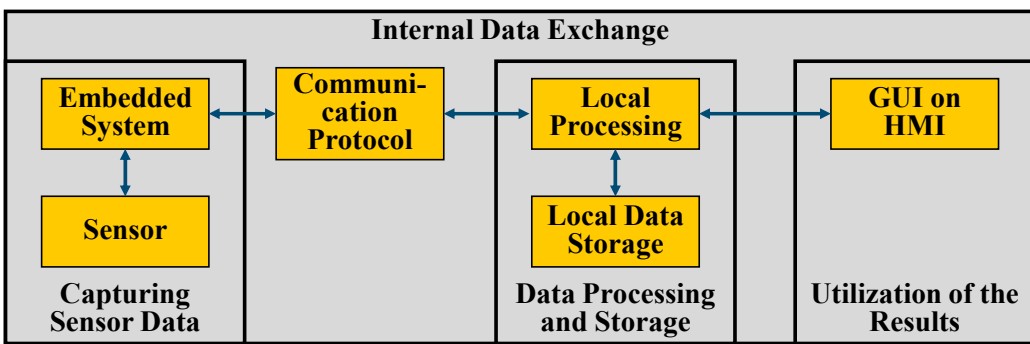

**Figure 7.** Schematic representation of the local architecture.

With local architecture, the computing power and storage capacity are to be created and provided on site. This allows the dimensioning to be adapted to individual requirements but is accompanied by an initial investment. If the system is to be expanded later and components replaced, further investment is required. Due to the fact that data are only processed within the local network, low latencies and fast real-time processing are possible [63,64]. Bandwidth limitations are not a problem in local networks. The fact that the data do not leave the local network reduces the vulnerability to cyberattacks [63,64] and offers a high degree of data sovereignty [65]. Furthermore, local architectures are not dependent on internet providers and are less susceptible to network disruptions. Fully decentralised, local computing is 14% less energy-intensive than fully centralised cloud computing [65,66].

### 4.2. Cloud Architecture

The basis for cloud architecture is cloud computing. This involves centralised computing services (e.g., storage, computing power) being made available via the Internet [56,63,67]. To this end, data are transferred from a local network to an external network via the internet so that data processing and storage takes place on servers there. As with the local architecture, the communication protocol, which defines the data transmission, is located adjacent to the CPS in which the sensor data are captured. The key difference here is that the data are not transmitted to processing units located in the local network but to processing units located on an external network, the eponymous cloud. The database, which is used for data storage, is also located on an external network (often the same one). The results are transmitted from the external network back to the local network, where the user can interact with an HMI via a GUI [52]. Figure 8 shows a schematic representation of the cloud architecture.

When using a cloud architecture, the existing hardware of the cloud provider is utilised. The bookable and usable computing and storage capacity can be dynamically adapted to the requirements, and performance peaks pose fewer problems [56]. The maximum achievable computing and storage capacities are usually very extensive [63]. Even if the use of cloud services incurs fewer or no initial investment costs [67], the booking of external services by providers incurs ongoing costs, which increase with growing performance requirements. Data transfer costs in particular can quickly become very expensive [63,68]. If large amounts of data have to be transferred, high latencies can occur when processing the data [56,63]. If a complete loss of connection to the external network occurs, this is accompanied by a loss of functionality of the Digital Twin [61]. Furthermore, the fact that the data leave the local network means that data security and data sovereignty cannot always be guaranteed.

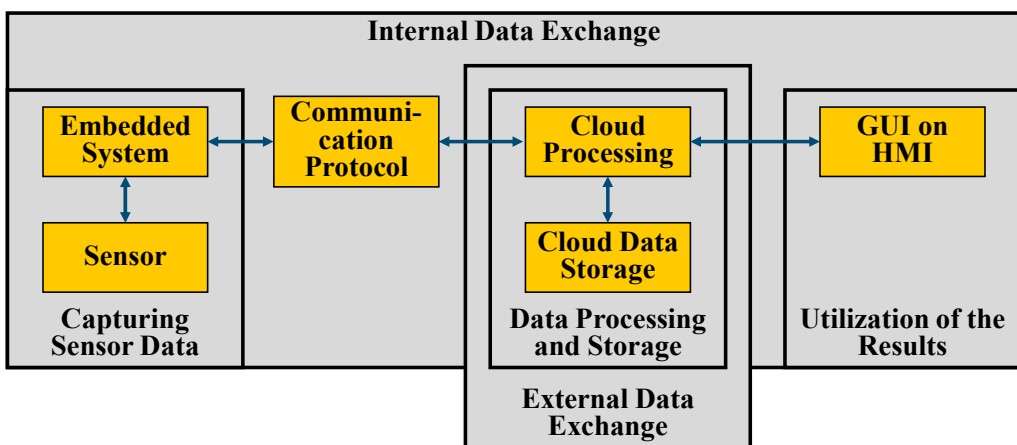

**Figure 8.** Schematic representation of the cloud architecture.

### 4.3. Hybrid Cloud–Edge Architecture

The hybrid cloud–edge architecture is a combination of the local and cloud architecture and data are processed both locally on the so-called edge device as well as in the cloud. The exact division is not absolutely defined and can be customised to the individual use case. Accordingly, the characteristics of the individual cloud and edge aspects are also variable. The distribution of computing and network infrastructure is also referred to as the edge-to-cloud continuum [69,70]. This also brings with it new challenges such as scalability, dynamicity and partitioning into smaller sub-clusters [69]. These challenges represent research areas in their own right and will not be discussed further in this article.

In the cloud–edge architecture, the communication protocol has two bidirectional connections that link the CPS both to the data processing and storage units in the local network and to those in the external network (cloud). One possible application scenario is the outsourcing of very compute-intensive operations to the cloud, while a few compute-intensive but possibly security-critical calculations are performed on the edge device [51,71]. An even more specific use case for this is training AI models in the cloud and then using them locally [56,64]. It is also possible to preprocess the data on the edge device and filter them or reduce them by extracting features such as mean values. This reduces the amount of data and therefore the bandwidth required to send the data to the cloud. Data storage can be realised both in the cloud and on a local server. In the same way that calculations are performed, a distinction can be made here between the amount of data and the criticality of the data. As with the edge architecture, local components can be combined on one end device. Due to the outsourcing of more computationally intensive processes to the cloud, this consolidation within the cloud edge architecture is a valid option not only for small systems. Similar to the other two architectures presented, the results are made available to the user via a GUI on an HMS. Figure 9 shows a schematic representation of the hybrid cloud–edge architecture.

The locally provided hardware and software are typically used to fulfil less computing- and storage-intensive tasks, which reduces the investment costs in this regard. The cloud is used for tasks with greater computing and storage requirements. This also allows flexible customisation to the task. Preprocessing on the decentralised local devices allows data reduction techniques to be used so that only data that contain new or relevant information content are transferred to the cloud. In this way, redundant data can be discarded and the volume of data to be transferred can be reduced [51,68,72]. In particular, this reduces costs resulting from data transfer rate and bandwidth requirements. Tasks that have high requirements in terms of real-time capability or data security and data integrity, as well as safety-critical functions that must not fail, can be executed on the local edge device. Functions that need to be accessible regardless of location can be located in the cloud.

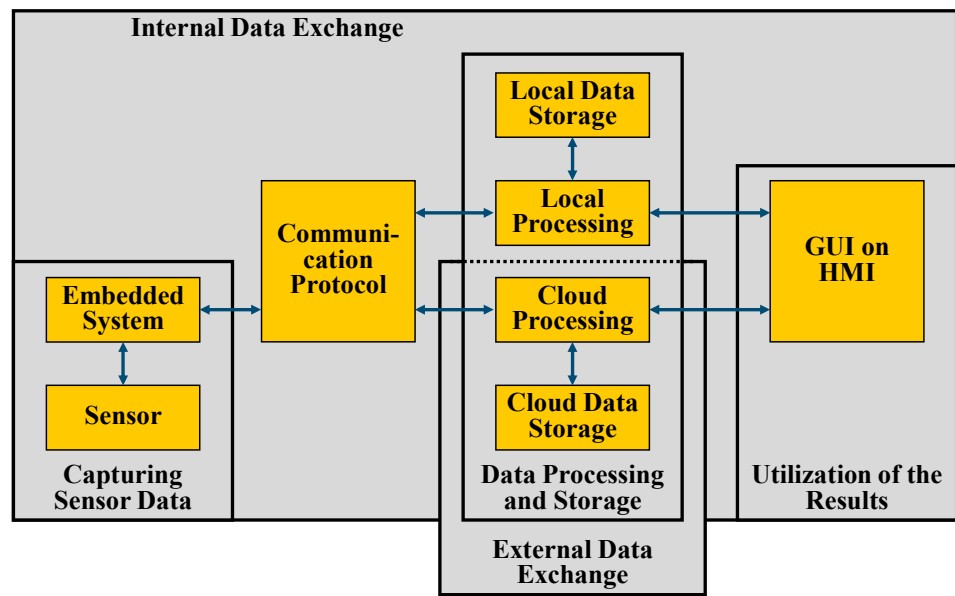

**Figure 9.** Schematic representation of the hybrid cloud–edge architecture.

**5. Procedure for the Creation of an IT Infrastructure for Digital Twins**

The aim of medium-level architecture is to reduce the complexity of developing an IT infrastructure for Digital Twins. To further address this goal, a procedure for the creation of an IT infrastructure is proposed below. On the one hand, these procedural steps are derived from the analysis of the RAMI 4.0 and the empirical use cases as well from knowledge gained during the derivation of the medium-level architecture. On the other hand, some aspects of the holistic approaches to the development of Digital Twins described in Section 2.4 are adopted. Although the authors of this article recommend following these steps, they are not mandatory. Figure 10 shows an overview of the steps, which are described in the following in more detail.

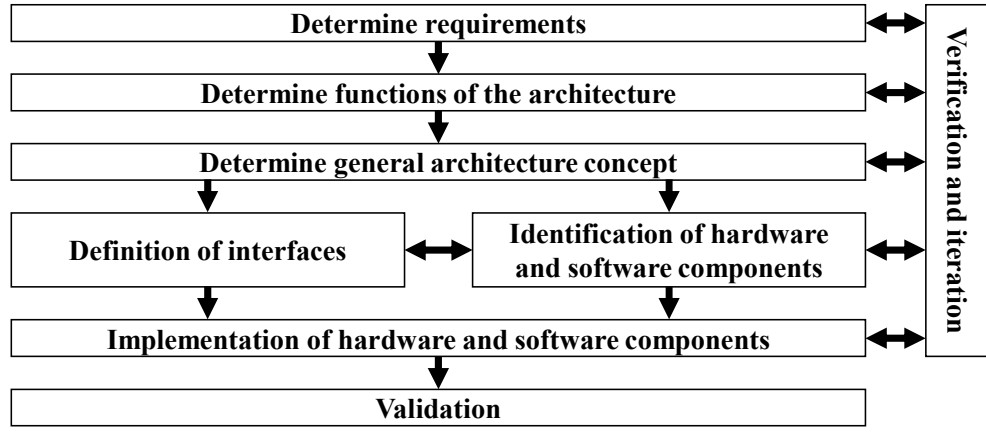

**Figure 10.** Process steps for creating an IT infrastructure for Digital Twins.

Determine requirements: The first step is to determine the requirements for the IT infrastructure. The use case and purpose of a Digital Twin and all relevant stakeholders must be taken into account, as well as economic aspects or potential business cases and technical requirements such as ones already mentioned above. These are particularly relevant for the following steps, which is why they are emphasised again here: computing power, real-time capability, data volumes, data security and data sovereignty, availability, accessibility, economic factors and energy consumption.

Determine functions of the architecture: The functionality of the IT infrastructure must also be defined. The functionality depends on the requirements placed on the IT infrastructure and therefore interacts very strongly with these. If a later expansion of the functional scope is considered, this should also be documented. In addition to the general requirements, security functions in particular must be taken into account. This can include the limitation of access. The targeted encryption of data is also a possible functionality of the architecture.

Determine a general architecture concept: The next step is to determine a general architecture that is capable of fulfilling the previously defined functions. For this purpose, the medium-level architectures are used. In particular, the requirements must be taken into account in order to decide between a local, cloud or hybrid cloud–edge architecture. The selected architecture represents a logical architecture and a solution for the tasks on a semi-abstract level.

Identification of hardware and software components: The selected medium-level architecture is then broken down into its necessary aspects, and specific components are defined. The components can consist of software or hardware components; often, a combination of both is necessary. These components serve to fulfil the tasks of the IT infrastructure at a more detailed level.

Definition of interfaces: Interfaces must be defined for data exchange between different components. Depending on the type, these interfaces can simultaneously consist of a physical interface (connectors, cables, wireless connection) or a digital interface in the form of a communication protocol (UPC UA, MQTT...) and data exchange format. Interfaces exist between the components of an architecture but also to adjacent components such as the models or the sensors. If several models are used that are to communicate with each other within the IT infrastructure, suitable interfaces must also be defined for this.

Implementation of hardware and software components: The hardware and software components identified in the previous step and their interfaces are then implemented. This includes on the one hand the acquirement and physical construction of the hardware and, on the other hand, the creation of program code to fulfil the software-side tasks. It may also be necessary to acquire additional services such as cloud computing power. On completion of this step, all components are implemented and integrated into an overall IT infrastructure system. In particular, possible security risks with regard to data security and security must be taken into account. Appropriate software components must be used for this purpose, and hardware restrictions may also be necessary.

Verification and iteration: During the implementation of all previous steps, constant verification and iteration must be carried out. This involves continuously checking whether the (interim) results achieved meet the requirements. The requirements are continuously supplemented on the basis of the knowledge gained during development. This means that various steps have to be run through several times until a satisfactory result can be achieved.

Validation: At the end of the creation of the IT infrastructure, the final result is validated. This involves checking not only whether the requirements have been met but also whether the right requirements have been set to meet the needs of the stakeholders and fulfil the purpose of the IT infrastructure.

## 6. Application Using the Example of an Industrial Gearbox

In the following, the results of the previous sections are demonstrated by applying them to a hypothetical example. This example is a two-stage industrial gearbox, which is part of a gearbox test rig at the Technical University of Darmstadt. A prototypical Digital Twin is being developed for this gearbox, for which a suitable IT infrastructure is required. The overarching use case of the prototype Digital Twin is the condition monitoring of the gearbox, in particular the critical components of the gears and roller bearings. The secondary purpose is to enable students to work with the Digital Twin, gain an understanding of it and expand it on a modular basis.

Determining requirements and defining the functions of the IT infrastructure: First, the requirements placed on the IT infrastructure are determined. For reasons of presentability, only a small selection of requirements will be discussed here. Due to the total number of sensors installed, which record operating data at a high sampling frequency, large amounts of data are generated simultaneously within a short period of time, which must be processed by the IT infrastructure. This requires the system to have a high write speed. At the same time, since no long-term measurements are carried out due to the prototype nature of the Digital Twin, the absolute data volumes are rather small in total, which also determines the requirements for moderate storage space. The models on which the Digital Twin is based are rather simple calculation and simulation models. Therefore, there are no hard requirements for the computing power. As the main users or editors of the Digital Twin are persons in the university context, good data accessibility as well as simple adaptability and modularisability of the system are important. Location-independent access from outside the shop floor is not necessary. Ongoing costs, such as external services or subscriptions, should be avoided.

Determine a general architecture concept: If the requirements are compared with the characteristics of the various medium-level architectures, the local architecture is the choice for this use case. The local architecture consists of data acquisition, local processing and data storage and an HMI with a GUI.

Identification and implementation of hardware and software components: Based on the general concept of the local architecture and the other requirements, suitable components were identified for the elements of the IT infrastructure and then implemented.

With regard to the capturing of sensor data, the existing system already provides boundary conditions, particularly on the hardware side. Sensors for temperature, vibration, speed and torque are already installed in or on the gearbox and can be used for data acquisition. Several HBM QuantumX MX840B universal measuring amplifiers are used as the interface. The manufacturer of the universal measuring amplifiers, HBM, also offers software for reading out the recorded measurement data. However, this software does not allow seamless further processing of the data by a Digital Twin. For this reason, a software solution must be developed utilising the HMB API, which records the measurement data and enables seamless further processing.

The main function of the Digital Twin, the calculation of damage conditions of the gears and bearings, is carried out using calculation models that were developed separately from the IT infrastructure and run on a Python (Version 03.12.0) application. This implementation is computationally efficient and does not place high demands on computing power. As part of an iterative process and in line with the other components, it was decided to provide the necessary computing power using a Lenovo ThinkPad T460 with a i5-6300U 2, 40 GHz processor and 8 gb of RAM.

No long runtimes are monitored as part of the prototype implementation of the Digital Twin. Despite the high sampling rate of the total of eight sensors, no large amounts of data are to be expected that need to be stored as part of the data storage. However, due to the high number of sensors, a fast write speed is required. An SSD with 240 gb is therefore selected. This is also installed in the ThinkPad T460, which provides the computing power. The data storage format is not subject to any special requirements due to the small amount of data. As part of the iterative process, a CSV file was chosen as this offers good synergies with the measurement software to be created.

The results of the calculations of the Digital Twin should provide the user with an initial overview of the damage and remaining service life of the gearing and ball bearings as quickly as possible. As the models are already running on a Python application, a graphical user interface in the form of a Python dashboard was chosen. This is to be displayed on an HMI, for which the 14-inch screen of the ThinkPad T460 is used.

Since all components except the measuring amplifiers are located within an existing system (ThinkPad T460), it is not necessary to consider the physical interface here. A

physical connection is required between the QuantumX and the ThinkPad T460, which is realised via an eight-pin LAN cable.

Verification and validation: The IT infrastructure created fulfils all the requirements that were originally set and which were added during the development process. The fact that the IT infrastructure is usable as described shows that the correct requirements were also set.

Figure 11 shows the result of the exemplary application on a two-stage industrial gearbox in the form of a schematic diagram. As in Figure 5, the software components are shown in blue and the hardware components in green. The models and sensors are shown without colour due to their special consideration parallel to the IT infrastructure.

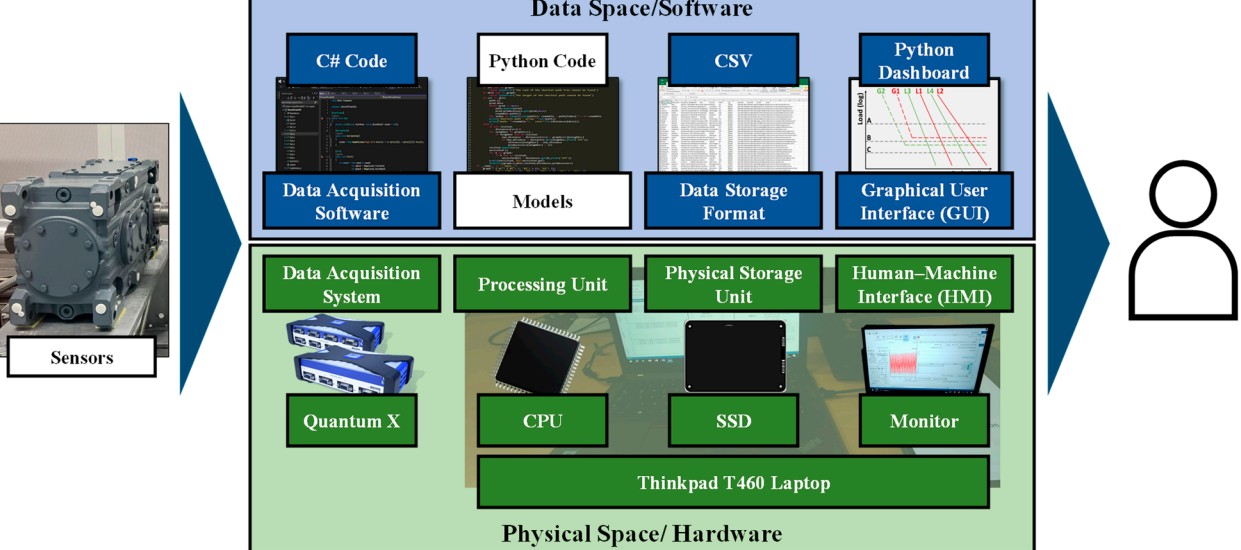

**Figure 11.** Schematic representation of the exemplary application on a two-stage industrial gearbox.

## 7. Conclusions, Critical Assessment and Outlook

This article presents an architecture for Digital Twins with a medium degree of abstraction, which is intended to create an IT infrastructure for a Digital Twin with less effort. For this purpose, the reference architecture for Industry 4.0 (RAMI 4.0) was used, and the Digital Twin concept was integrated in it. This enabled the tasks of the IT infrastructure to be identified and research needs to be determined. Furthermore, key requirements for an IT infrastructure were examined. These requirements and their application-specific characteristics can be met with the help of many different architectures, which can be summarised into three groups: local architecture, cloud architecture and hybrid cloud–edge architecture, whereby mixed forms are possible. This functional decomposition and its transfer into one of these three architecture groups with the help of requirements represents the medium-level architecture. Thus, it can be used to close the gap between the abstract reference architectures such as RAMI 4.0 and a specific implementation (RQ1). With this contribution, a sequence of steps was proposed that a developer can follow to create an IT infrastructure (RQ2).

When assessing the results presented, it is important to consider the background of the authors of this article. While the creation of the IT infrastructure of a Digital Twin is an interdisciplinary task, the focus is very much on software engineering. This is particularly true for the later stages of the procedure presented. The authors of this article all have a background in mechanical engineering, which is why the goal formulated at the beginning of this article should be emphasised once again; the development of an IT infrastructure for Digital Twins is to be simplified by creating a uniform understanding in interdisciplinary development teams. There is no ambition to be able to train experienced software engineers. In the dissemination of the results of this contribution, both in the

medium-level architecture and in the step-by-step procedure, various sources of information were taken into account and efforts were made to consider different perspectives and to cover as complete a picture as possible. The aim was to reduce the risk of blind spots in the analysis. Despite these efforts, it must be emphasised that the results were mainly derived either directly or indirectly from the literature. For these reasons, it cannot be said with certainty that the concepts are effective or efficient. Verification has not yet taken place and is urgently sought.

To develop a Digital Twin, sensors and models of the physical twin must be considered additionally to the IT infrastructure. There are already approaches for a systematic approach in the literature [4,50]. However, it is rarely expedient to view these three domains independently as they are interacting. In order to develop a Digital Twin, all three domains and, in particular, the interactions between these must be taken into account. This requires a holistic approach that combines the process steps of IT infrastructure, sensor technology and modelling. This holistic approach or methodology is the subject of further research.

There is also a need for further research into the IT infrastructure of Digital Twins individually. This includes, for example, concrete steps for selecting and evaluating specific components for creating the infrastructure. In particular, this can include the selection of suitable cloud solutions.

Digital Twins are a key component of Industry 4.0, and categorisation in RAMI 4.0 only underlines this. The conscious consideration of humans in the system and the humanisation of industry are currently paving the path to Industry 5.0 [73,74]. In the long term, the use of Digital Twins in this expanded context and the associated opportunities and risks should be considered. This represents a further research direction.

**Author Contributions:** Conceptualization, M.F. and M.K.; methodology, M.F.; writing—original draft preparation, M.F., M.K. and F.W.; writing—review and editing, F.W., S.G., S.W. and E.K.; visualization, M.F. and M.K; funding acquisition, S.W. and E.K. All authors have read and agreed to the published version of the manuscript.

**Funding:** This project is supported by the Federal Ministry for Economic Affairs and Climate Action (BMWK) on the basis of a decision by the German Bundestag. It is part of the IGF Project 22467 BG (FVA 889 II Digital Twin II), in collaboration with the Forschungsvereinigung Antriebstechnik (FVA) e.V.

Supported by:

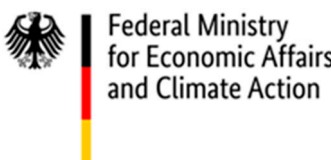

**Data Availability Statement:** Data are contained within the article.

**Conflicts of Interest:** The authors declare no conflicts of interest.

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
