# Peer review of "Medium-Level Architectures for Digital Twins: Bridging Conceptual Reference Architectures to Practical Implementation in Cloud, Edge and Cloud–Edge Deployments"

_electronics, doi:10.3390/electronics13071373_

Round 1
Reviewer 1 Report
Comments and Suggestions for Authors
“Medium-Level Architectures for Digital Twins” seeks to fill a gap in the literature focusing on the creation of IT infrastructure necessary for digital twins. The authors note a high degree of abstraction in most literature and take an approach that could be likened to a chapter aimed at guiding graduate students. This is both the novelty of the paper and what could most easily open it to criticism as it is not the traditional research paper. The paper sufficiently addresses its goals and is well structured with a compressive “Fundamentals” section after the introduction. The section is better detailed than most articles and again highlights the paper’s value as a functional resource.
I do have a couple minor suggestions.
RAMI 4.0 is discussed along with mentioning other reference architectures. A more detailed analysis on the strengths and weaknesses of other architectures in relation to the proposed approach would provide greater context.
Also, security is an important consideration for companies and addressing issues such as encryption and access control could strengthen the paper.
Author Response
Thank you for your valuable feedback and the opportunity to improve the quality of the article. We have implemented your feedback as follows: Following the list of architectures, the characteristics and deficits of these are described. The requirement for data security was supplemented by aspects of encryption and access control. In addition, the aspect of data security has been emphasised separately in the step-by-step procedure.
Reviewer 2 Report
Comments and Suggestions for Authors
electronics-2947342
Review, 22 March 2024
Dear Authors,
Your paper is very good, as a reviewer I have only minor comments, here are:
Note 1: To the title of your publication add that the analysis was made on the basis of the structure of the RAMI model, because in the keywords it is RAMI and in the title it is not.
Note 2: In sections (subsections) titles, do not use color names equivalent to RAMI models, use the words "yellow", "blue" etc. in the text description, not in section titles.
Note 3: The RAMI model is for I 4. 0, did you not miss to write at the end of the work that now the concept of Industry 5. 0 is popularized. Many authors, as well as I also focus on humanization in Industry 5. 0 , but after all the base is technology I 4. 0. , so your model is also useful there.
Best wishes
Reviewer
Author Response
Thank you for your valuable feedback and the opportunity to improve the quality of the article. We have implemented your feedback as follows: The description of the RAMI 4.0 colours (yellow, blue, red) has been removed from the headings of the subsections. In the Outlook chapter, the results were placed in the Industry 4.0 concept and an outlook on Industry 5.0 was given. The proposed change to the title was not implemented. The reason for this is that the RAMI 4.0 is only one of many sources used to create the results of the article. Including RAMI 4.0 in the title would give it too much focus.
Reviewer 3 Report
Comments and Suggestions for Authors
The following comments to be addressed:
The authors must perform the experiments and provide results to justify the validity and superiority of the proposed architecture.
Author Response
Thank you for your feedback and the opportunity to improve the quality of the article. Our contribution focuses on the question of what the process of creating IT infrastructures for digital twins might look like. The aim is always to create the most appropriate infrastructure for the use case. However, the aim of this article is not to present a universally best solution. The application example at the end of the article is primarily just an illustration of the stepwise approach. For this reason, we do not believe it is appropriate to conduct experiments with the aim of demonstrating the superiority of this architecture, since this was never the goal.
Round 2
Reviewer 3 Report
Comments and Suggestions for Authors
The following comments to be addressed before considering to publish.
1. The title of the paper too long. Make it two lines.
2. Cite most recent works on Cloud-edge continuum. For example, Exploring the potential of distributed computing continuum systems
3. Advantages and limitations of the proposed work are added. Also, provide how does this work can be extended further in the near future.
Author Response
Thank you for your valuable feedback and the opportunity to improve the quality of the article.
The section on hybrid cloud-edge architecture was supplemented by introducing the cloud-edge continuum and the associated challenges. The challenges are not the focus of the article; instead, reference is made to the relevant literature.
Advantages and limitations of the proposed work are discussed in section 7 ("Conclusions, critical assessment and outlook"). In doing so, it is discussed how the results can be utilised and how they have answered the research questions. In addition, the limitations, especially those resulting from the authors' engineering background, were also discussed. In addition, future extensions were considered by combining the approach with sensor technology and modelling to create a holistic approach. Furthermore, an outlook on the extension of the results through selection processes of more concrete components is given.
We would like to avoid changing the title. In our view, it is hardly possible to achieve the same meaning with a significantly shorter title.